# Development of Impact-Echo Multitransducer Device for Automated Concrete Homogeneity Assessment

**DOI:** 10.3390/ma14092144

**Published:** 2021-04-23

**Authors:** Bartłomiej Sawicki, Tomasz Piotrowski, Andrzej Garbacz

**Affiliations:** 1Department of Building Materials Engineering, Faculty of Civil Engineering, Warsaw University of Technology, Al. Armii Ludowej 16, 00-637 Warsaw, Poland; bartek.sawicki@epfl.ch (B.S.); a.garbacz@il.pw.edu.pl (A.G.); 2Laboratory for Maintenance and Safety of Structures, Civil Engineering Institute, Swiss Federal Institute of Technology Lausanne (EPFL), EPFL ENAC IIC MCS Station 18, CH-1015 Lausanne, Switzerland

**Keywords:** concrete, NDT, finite element method, experiment, automated inspection, frequency domain, validation

## Abstract

A combination of multiple nondestructive testing (NDT) methods speeds up the assessment of concrete and increases the precision. This is why the UIR-Scanner was developed at Warsaw University of Technology. The scanner uses an Impact-Echo (IE) method with a unique arrangement of multiple transducers. This paper presents the development of the IE module using numerical models validated with experimental testing. It was found that rectangular arrangement of four transducers with the impactor in the middle is optimal for quick scanning of area for faults and discontinuities, changing the method from punctual to volumetric. A numerical study of void detectability depending on its position with respect to the IE module is discussed as well. After confirmation of the findings of models using experimental tests, the module was implemented into the scanner.

## 1. Introduction

Nondestructive techniques (NDT) for examination of structures without their degradation are within the essential field of engineering art development [1,2,3]. Each of them has its own characteristics in terms of accuracy, rapidity and kind of faults they can detect. This is why a combination of multiple techniques is of interest. Usage of different NDTs brings the problem of results’ correlation and necessity of execution of multiple measurements. Nevertheless, it speeds up the assessment and this is why merging multiple techniques into one device is currently an emerging trend [1,2,4]. 

According to this philosophy, a scanner for nondestructive testing of concrete slabs was developed at Warsaw University of Technology [5]. The UIR-Scanner merges complementary NDT methods: **U**ltrasonic pulse velocity (UPV), **I**mpact-echo (IE) and ground penetrating **R**adar (GPR) to deliver reliable results in an automated and user-friendly manner.

This paper presents the conceptual development of an Impact-Echo module for the scanner. Its task is to detect anomalies in the tested region, which may indicate presence of a fault. Once areas with the anomalies are marked, they can be tested in more detailed ways to obtain the precise location of fault. The detectability criterion should be quantitative allowing automated analysis by software.

## 2. Impact-Echo Method

The Impact-Echo (IE) method is a nondestructive technique invented at the turn of ‘70 s and ‘80 s for the testing of solid concrete and multilayer [6,7,8,9] elements. This method is based on application of impact of a steel ball at the surface of the structure to evoke an elastic stress wave, which later reflects and is recorded with a piezoelectric transducer. The received signal in time domain is changed to frequency domain by Fast Fourier Transform (FFT) and then it is analyzed and interpreted to obtain the thickness of the element and, possibly, the depth of reinforcement and flaws (Figure 1).

A hammer or, most commonly, a steel ball is used as an impactor. The diameter of the ball governs the contact time [7,10], and thus, the frequency content of produced wave. The expected resonance frequency of the tested element should be included in the produced frequency range [11]. Usually, steel balls of diameters 4 mm to 30 mm are used, which allows for the testing of the concrete element of thickness up to 1 m [12]. Another wave source, like an air gun [13], can be used as well.

An elastic stress wave produced with an impactor contains three components: (I) Rayleigh (R) surface wave, (II) Shear (S) wave and (III) Compression (P) wave [9,11]. The P wave travels the medium and reflects from lower and upper faces of the element, as well as from any faults or borders of underlying layers. This phenomenon is caused by a difference in the mechanical impendences of materials in the element, and especially between air and concrete [7,12]. The periodicity of echoes produces resonance frequencies.

These periodic reflections are recorded with a piezoelectric transducer. A device with a large bandwidth between 2 kHz and 50 kHz is used. The abovementioned frequencies correspond to average thickness of concrete element between 5 cm and 1 m [12]. Another technique, e.g., laser interferometry [14] or microphone [15,16], can be used in the air-coupled approach.

For sake of interpretation, the registered signals are usually presented in frequency domain using fast Fourier transform [17]. The use of frequency domain allows for observation of multiple reflections of wave between upper and lower face of the element, making the measurement insensitive to the distance between impactor and receiver [9]. The wavelet analysis can be used in the analysis as well [18].

The thickness of the element can be computed knowing both: (I) the resonance frequency of P wave, and thus the travelling time, and (II) the velocity of P wave. The velocity of the P wave can be either measured [12] or obtained empirically [6]. A similar procedure is adopted for any layers’ interfaces or faults present in the tested element, visible as peaks in the frequency spectrum (Figure 1).

IE is very similar to the ultrasonic method, but two significant differences are present: (I) ultrasonic waves are linear, which enables measurement only in line of excitation, while stress-waves in the IE propagate spherically; and (II) the frequency of waves in the IE method is lower, and therefore, heterogeneity of concrete has neglectable influence on the results. Additionally, the spherical wave propagation allows for separation of the excitor and receiver, making IE more flexible and capable of testing bigger areas and volumes, thus making it faster in use.

Beside measurement of the thickness of element, the Impact-Echo method has been used to detect: flaws [8], cracks [19], voids and debondings [7], honeycombing [20], delaminations and quality of interfaces [21,22] or voids in grouted tendon ducts [23].

The numerical finite element methods are important to ensure the IE development [8]. They have been used extensively to simulate different types of flaws and understand the results obtained with IE testing. From point of view of this paper, the sensitivity of IE to size of faults [23] and fusion of results from multiple sensors [24] are especially important.

Several trials with automated NDT using IE [25,26] and its fusion with other techniques [27] has been undertaken before. The commercial devices were also developed, e.g., Olson Engineering INC-Bridge Deck Scanner (BDS) [28] or BAM NDT Stepper [29].

## 3. Preparation and Validation of Finite Element Model

To obtain reliable results from the numerical model, verification and validation is needed [30]. The parametrical studies by variating modulus of elasticity, Poisson ratio and density of concrete to change wave velocity, as well as finite element size, were conducted. The results were verified against analytical solution as described below and good agreement was found. For sake of brevity, detailed results are not presented here, and are available in [31].

### 3.1. Experiment for Validation

To validate the numerical model, an experiment on solid concrete slab was used. The slab of dimensions 50 cm × 50 cm × 7 cm was casted using C20/25 class concrete according to Eurocode and supported on 5 cm by 5 cm washers in four corners. After curing, 30 IE tests using commercial DOCter^®^ Mark IV commercial device [32] were executed. A 2 mm ball was used to evoke the wave in a distance of 5 cm from the receiver. All the measurements show the thickness of 70 mm, so no internal faults were present. The signal was probed approximately every 2 × 10^−3^ ms, and the total length of the obtained signal was 1.9 ms per measurement. All the results were recorded and saved for further comparison with the numerical model.

### 3.2. Numerical Model

The abovementioned slab was modeled in a commercial program LS-DYNA^®^ [33] and is presented in Figure 2a. A three-dimensional model was built using 8 nodes hexahedron 7 mm × 7 mm constant stress solid elements of type ELFORM 1. This type of element, with a single integration point, was chosen for better computational efficiency considering the relatively simple stress-state and shape of the modeled structure. Viscous control of hourglass (type 1) was chosen as it is recommended for higher velocity waves and is computationally the cheapest. Considering the relatively low size of mesh, good results were obtained with this simple modeling method and explicit integration.

Vertical displacements were constrained in corners on the lower face of the plate. The material was modeled as perfectly elastic, with modulus of elasticity 38 GPa, density 2400 kg/m^3^ and Poisson ratio 0.2, corresponding to C20/25 concrete [10]. The excitement was applied by force with sine function variability of 2 kN. Such variability is commonly used and its magnitude does not influence the results thanks to the elastic material model [6,9]. The excitation impulse lasted for 8.4 × 10^−6^ s, corresponding to a 2 mm ball [6]. Vertical displacement of the node in location of the transducer was registered for 2 ms, with a time step of 1.4 × 10^−3^ ms. 

### 3.3. Signal Processing

To validate the model, two signals collected in different ways needed to be compared. During the experiment, variation of current voltage due to pressure waves acting on piezoelectric in transducer was registered, while the direct displacement of node could be obtained from the model. This is why a proper filtering is important [17].

For validation, the surface (Rayleigh) wave was removed by cutting the first 0.5 ms of signal. Then, the Butterworth filter was applied for frequencies outside ½*f–*2*f* window, where *f* is the frequency expected due to plate thickness. The filtering was needed mostly due to numerical noise and eigenvibrations present in numerical results, as well as high frequency noise due to the concrete inhomogeneity in slabs. Then, the Fast Fourier Transform (FFT) was used to present and compare the frequency spectrum.

### 3.4. Results

Thickness of the tested plate, and depth of possible fault, can be found on the basis of the passage time of the pressure wave through the material. Since the wave is reflected multiple times from the bottom and top surfaces, frequency of the wave passage is used. The velocity of wave can be obtained from the empirical Equation (1) [6]:(1)$$C_{p}=\sqrt{\frac{E(1-\nu )}{\rho (1+\nu )(1-2\nu )}}$$
where *E* stands for Young’s modulus of elasticity of concrete, ρ for density of concrete and ν is Poisson’s ratio. This velocity is, however, dependent on the element’s shape; thus, for plates, an apparent velocity should be used [8,9], leading to the Equation (2):(2)$$T=\frac{C_{pp}}{2f}$$
where *f* stands for frequency of signal, *T* for thickness (depth) of element and *C_pp_ =* 0.96*C_p_* is the reduced, apparent wave velocity for the concrete plate with *υ* = 0.2 due to resonance and the creation of Lamb waves [34,35]. 

A good correlation between registered signals and their frequency spectra was obtained. The dominating frequency from experiment is 27.40 kHz, while from the model it is 27.19 kHz. With *C_pp_* = 3830 m/s, the thickness of 70.0 mm and 70.4 mm can be calculated, respectively.

Thus, the model was validated and good agreement with the experiment was found.

## 4. Experimental Studies on a Large Span Floor

The bigger the spacing between impactor and transducer, the larger the distance that wave travels in the material. Because of this, the area tested at once can be increased, reducing the time needed for the whole slab or floor, and IE can be used to test a certain volume of concrete at once. This is why balance between the reliability of results and the distance between impactor and receiver needed to be found.

An experiment was prepared to investigate the maximum distance between the impact and read-out points at which results remain clear and reliable. Because the circular wave reflects not only from faces of the element but from other boundaries as well, interference may influence the results. To avoid this, the large span 40 cm thick concrete floor was chosen. The span-to-depth ratio was big enough to considered it as an element of infinite area.

Measurements were taken in the three directions (each 45°) to avoid influence of reinforcement. The distance between points of impact and signal reception varied from 10 cm to 300 cm. A 12 mm impactor was used according to the geometry of slab and detection ability as presented by Carino [9]. Each measurement was repeated three times.

Measurements show that the maximum distance between impactor and receiver giving reliable results is around 90 cm and above this distance, the wave reaching transducer was too weak (Figure 3). It has to be stressed that energy produced by the smaller ball is lower, and thus, this distance would decrease. This result can be influenced by the type of the concrete used and serves merely as estimation of possibility of change Impact-Echo from punctual into volumetric method. Taking into account the expected range of thicknesses of slabs to be tested with the device under design, as well as convenience of its use, a square 60 cm × 60 cm was assumed as a maximal testing area. Furthermore, it was concluded that an automated impact procedure should be implemented to make uniform the subsequent readouts. The exact results and discussion can be found in [31].

## 5. Finite Element Model of Plate with Voids

Knowing the maximum dimensions of the IE module that could be incorporated into the automated scanner combining different NDT techniques [5], the best layout was pursued. The IE module should allow for testing in an automated way the biggest possible area at once. At this stage, the presence or lack of fault or void should be determined. The depth and exact location within the tested area can be found using more precise testing later. In this way, the large area of slab can be scanned in a quicker manner, showing the points of possible further interest.

### 5.1. Numerical Model Modification

It was decided to model the plate for further investigation as an infinitely large slab. The purpose of this was to avoid possible reflections from the boundaries of the element. To do this, a square plate 90 cm × 90 cm × 20 cm was modeled with further extension to obtain a circular external surface (Figure 2b). To this side surface, nonreflective boundary conditions were assigned. The vertical supports were assigned at the mid-span on the whole perimeter.

The void was modeled by removal of elements in the box 20 cm × 20 cm × 2 cm at mid-depth (Figure 4). The expected peaks, calculated with formula (2), are presented in Table 1, respectively, for the top and bottom of the void, and for the thickness of whole element. Since the element’s size, material properties and finite elements type remained the same as in validated model, no additional verification or validation was needed.

### 5.2. Possible Linear Layouts

The standard IE layout consist of one impactor and one receiver. If the void or fault is below one of them or between them, it can be detected. 

The first considered layout was one receiver and one impactor spaced by 40 cm, moving along the line perpendicular to them. This way, subsequent cross-sections of slab can be examined. However, to obtain the necessary resolution, many measurements were needed (e.g., every 10 cm), leading to a time-consuming procedure.

To cover a bigger area at once, a set of four impactor–receiver pairs each spaced at 10 cm, covering an area of 30 cm by 40 cm, can be used. All the impactors would be released at same time. However, due to wave interference, the readouts were difficult to analyze, and this approach was given up. Furthermore, the release of impactors at exactly the same time could be problematic.

The third layout would be a modification of the above. Instead of using four impactors, only one could be used. However, the distance between the impactor and each of the receivers would not be equal and analysis of such a signal would be difficult.

On the basis of the numerical models of the three abovementioned arrangements, it was decided that for the proper layout, the distance between each of receivers and impactor should be equal. Furthermore, to avoid interference of waves, only one impactor should be used for the IE module.

### 5.3. Square Layout

As mentioned above, it was decided that the distance between impactor and all transducers should be equal for sake of simplicity of comparison. Although the maximum distance between points of impact and read-out was found to be 90 cm, the IE module should fit within the UIR integrated scanner. Therefore, the distances were reduced. The transducer layout should allow diagnostics of the largest possible area at once. This is why the square scheme was investigated with piezoelectric receivers in points 1 to 4 (Figure 5). Then, the applicability of this layout with different positioning of void was modeled.

### 5.4. Signal Processing

The signal was processed in similar way as in the experimental validation (see Section 3.3). Filtering out the frequencies outside of ½ *f*–4 *f* was performed to exclude the eigenvibrations of plate and numerical noise, but to keep the high-frequency waves due to reflections from the void. The Raleigh wave was removed by cutting out first 0.5 ms of signal.

## 6. Results of Simulations

The results are compared in both qualitative and quantitative ways. The qualitative way is by comparison of frequency spectra and peaks that they reveal. This process can be automatized in the scanner. The quantitative ways used in this paper are: (I) integration of the frequency spectra, and (II) calculation of the root mean square (RMS) of the frequency spectra. The outcome of the comparison should be whether some anomaly in the measurement results can be spotted, and thus, if the void can be detected and indicate that there exists a need for further testing of the area.

### 6.1. Solid Plate (No Void) vs. Plate with Void

First, the comparison between solid plate and plate with void below the impactor and transducer was done. 

In the case of the solid plate, the peak due to thickness (9.95 kHz) was clearly visible for all sensors (Figure 6). The second highest peak (18.0 kHz) lied the around half-depth (10.4 cm) and might be the result of wave interference. The integral and RMS values (Table 2) were slightly different, possibly due to numerical noise. However, the obtained variation remained below 3%. Thus, if the absolute value of variation (Δ) of both integral and RMS is outside of 10% interval of the Reference Value (RV), the fault is detected. If only one value is outside of the interval, it indicates warning. The RV was taken as the mean value of integrals and RMS from a solid plate. 

When the void was located below impact point (Figure 7, Table 3), a big difference could be noticed from the previous case in the value of the integral and RMS. The values were smaller as the energy was largely dumped by discontinuity. Additional peaks for frequencies about 22.4 kHz were visible in the frequency spectrum, which allowed for detection of the void in the investigated area (Figure 8). 

If the void was located only under one transducer (Figure 9), it was highly affected by dumping, as visible in Table 4. However, the sensor No.3 was surprisingly slightly dumped as well, possibly by interference with wave reflected from edge of the fault. In frequency spectra (Figure 10) for test point 1, the decrease in magnitude of peak due to plate thickness (9.95 kHz) was clear, as well as the additional peak due to the void (22 kHz).

### 6.2. Edge of Void below Impactor

Another interesting case is the location of edge of the void directly below the impact point (Figure 11). While comparing the integrals and RMS values (Table 5) or frequency spectra obtained here and for the solid plate, a reduction of energy delivered to transducers was visible. It is most evident for sensors close to the void (i.e., 1 and 4, Figure 12). Thus, the void can be detected and its approximated location can be given, but not the depth.

### 6.3. Void Moving Away from Impactor

One more situation is the void moving away from the impact point (or, rather, when the scanner is moving away from the void, (Figure 13 and Figure 14). Although the frequency spectra (Figure 15 and Figure 16) did not precisely reveal the depth of the void, its location could be determined. Not only did the transducer closest to the fault reveal the biggest dumping of energy (Table 6 and Table 7) as described previously, but also a linear relation between received energy and void position could be found. In the Figure 17, the variation of RMS values with respect to reference value of solid slab are presented for three cases, as well as the linear fit quality.

### 6.4. Summary

Simulations presented above clearly show that with a square set of 4 transducers and 1 impact point, the impact-echo method turns into volumetric, and a void can be detected if it lays within the area covered by the device. Its approximated position can be given in any case. However, the depth can be found only if it lays below the impact point or under one of the sensors. However, this precision is sufficient and thus, the proposed scheme can be adopted in UIR-Scanner. 

The sensitivity of setup regarding minimum detectable fault was not analyzed. It can be expected that ability of detection is restricted by limits of impact-echo method [21], e.g., ratio of minimal lateral dimension of void to stress wavelength (impactor size) and element’s depth. Furthermore, the numerical model presents an idealized medium, which is not the case in real application. Therefore, during the implementation of proposed layout into UIR-Scanner, the limitations and problems that are inherent in the chosen NDT method need to be borne in mind [36], i.e., (1) reflection coefficient R between fault and medium needs to be large enough; (2) possible multiple reflections, e.g., due to reinforcement, can impair obtained results; (3) source of external vibrations and difference in vibration modes between types of plates can influence results.

## 7. Experimental Validation of Adopted Layout

Before implementation of an IE module in the UIR-Scanner, the layout was experimentally validated on a concrete slab using DOCter Mark IV transducers. The 50 cm thick slab of C30/37 concrete was used. At a depth of around 5 cm, a void was modeled by foamed polystyrene plate, 5 cm × 5 cm × 2 cm.

Measurements were executed with one transducer and one 2 mm impactor. The transducer was subsequently moved between positions 1 and 4, and the impact was applied in the center of setup (Figure 5). Two cases were considered—with no void, and with void below the impact point. The measurement was repeated 7 times at each measurement location (measurement A to G).

For obtained frequency spectra, Integral and RMS were calculated in range 0 Hz–70 Hz, where 70 Hz corresponds to twice the frequency due to the mid-depth of the void according to formula (2). The results for each measurement are presented in Table 8, Table 9, Table 10 and Table 11.

For each measurement, the elastic wave was induced by separate impact and transducer was lifted and pressed again every time. Therefore, the measurements are uncorrelated. To analyze the results, and eliminate isolated events of different impact energy or imperfect contact of transducer with surface, average values from seven measurements (A to G) were calculated and presented in Table 12 and Table 13 for slab without and with the void, respectively. The same criteria of detection as for numerical model were used; therefore, Δ above 10% for one of indicators (Integral of RMS) results in warning and for both indicators results in detection.

Despite relatively large discrepancies between individual measurements due to manual excitation of elastic stress-wave, average values clearly indicate presence of the void. The results confirm the need for automated excitation and simultaneous measurement with all four transducers. This experiment confirmed applicability of IE as a volumetric method. Therefore, the numerical modeling results were validated, and the IE module was implemented into the UIR-scanner (Figure 18 and Figure 19) and issued for further sensitivity studies.

## 8. Conclusions

The aim of research presented in this paper was to find the optimal design of an Impact-Echo module for automated detection of horizontal faults in concrete slabs. The IE module was to be incorporated into integrated UIR-Scanner combining three nondestructive methods. The IE module should determine possible locations of faults in the slab in the rapid, volumetric manner. Then, in indicated locations, further tests can be conducted to confirm the outcome.

The following recommendations were issued: (1) the impact should be automated to allow comparison of magnitude of frequency spectra between the measurements; (2) layout with four transducers at corners of square 60 cm by 60 cm and one impact point in the middle should be adopted as it allows the most efficient testing; (3) clear rules for signal analysis and interpretation, with use of frequency spectra’s integral and RMS, are established. They can be further introduced in the software of the scanner to automatize the procedure of detection. This layout was adopted into the UIR-Scanner, validated and issued for further experimental testing. 

The current research has clearly presented that relatively simple Impact-Echo method, usually used for punctual measurement of element’s thickness or detection of faults, can be modified into volumetric method by use of multiple transducers. Despite loss of precision and ability to localize the fault, this approach allows for much faster scanning of structures and opens new possibilities of application of the IE method for nondestructive testing of concrete.

## Figures and Tables

**Figure 1 materials-14-02144-f001:**
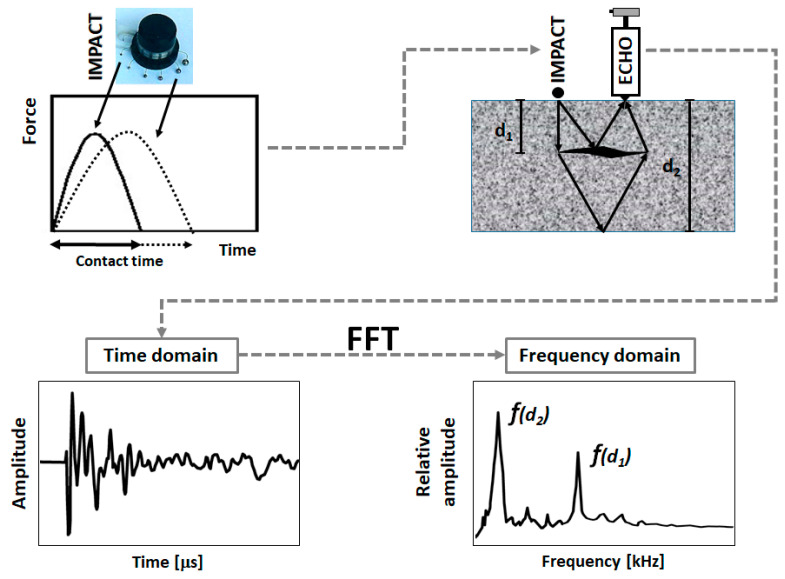
Scheme of impact-echo method with an example of waveform time-domain spectrum and corresponding frequency spectrum when defect in concrete is observed.

**Figure 2 materials-14-02144-f002:**
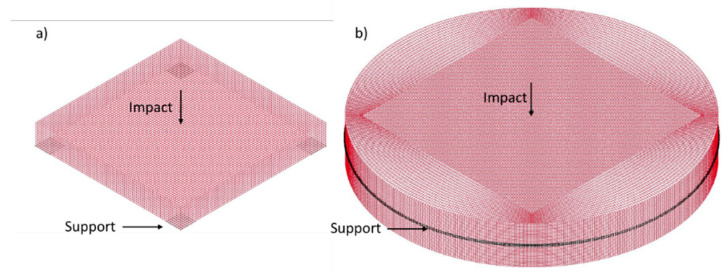
Isometric view of Finite Element Models (**a**) model of 50 cm × 50 cm × 7 cm plate for validation; (**b**) model of infinitely large slab for layout testing.

**Figure 3 materials-14-02144-f003:**
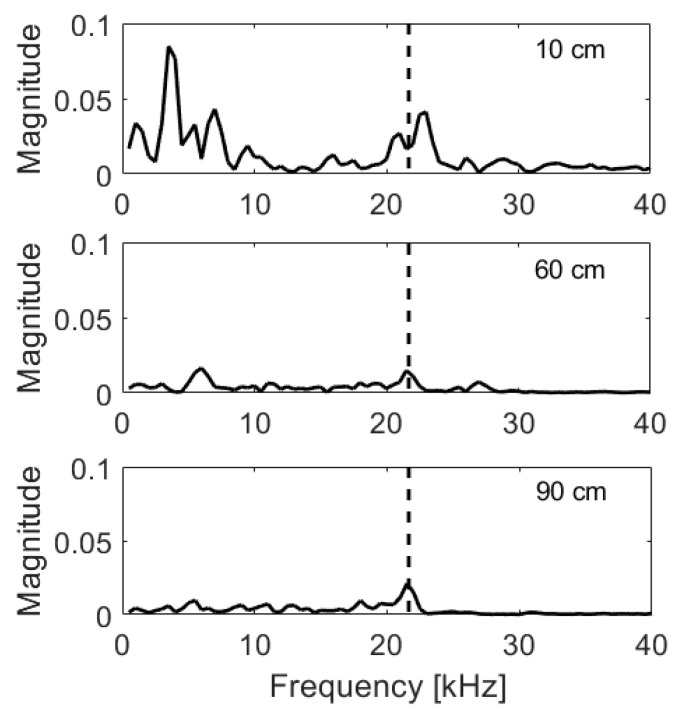
Frequency spectra of measurements taken at a large span floor in distance of 10 cm, 60 cm and 90 cm from impactor; frequency due to thickness marked with dashed line.

**Figure 4 materials-14-02144-f004:**
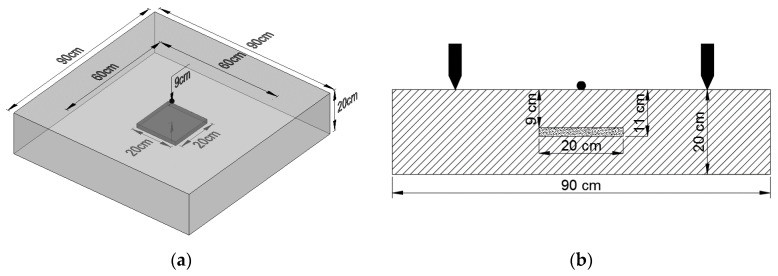
The plate with void: (**a**) view and (**b**) section.

**Figure 5 materials-14-02144-f005:**
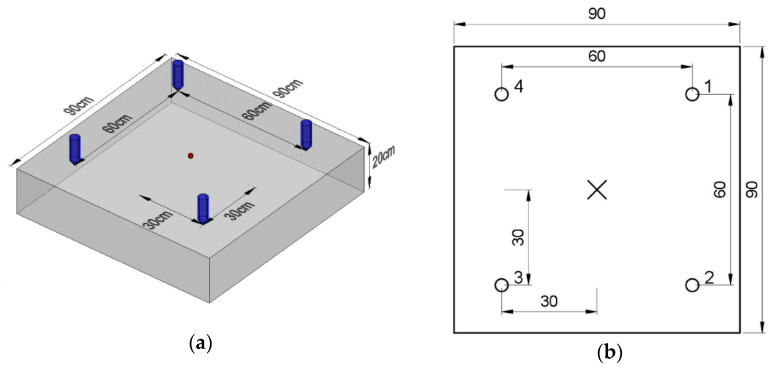
Square measurement system layout: (**a**) isometric and (**b**) top views.

**Figure 6 materials-14-02144-f006:**
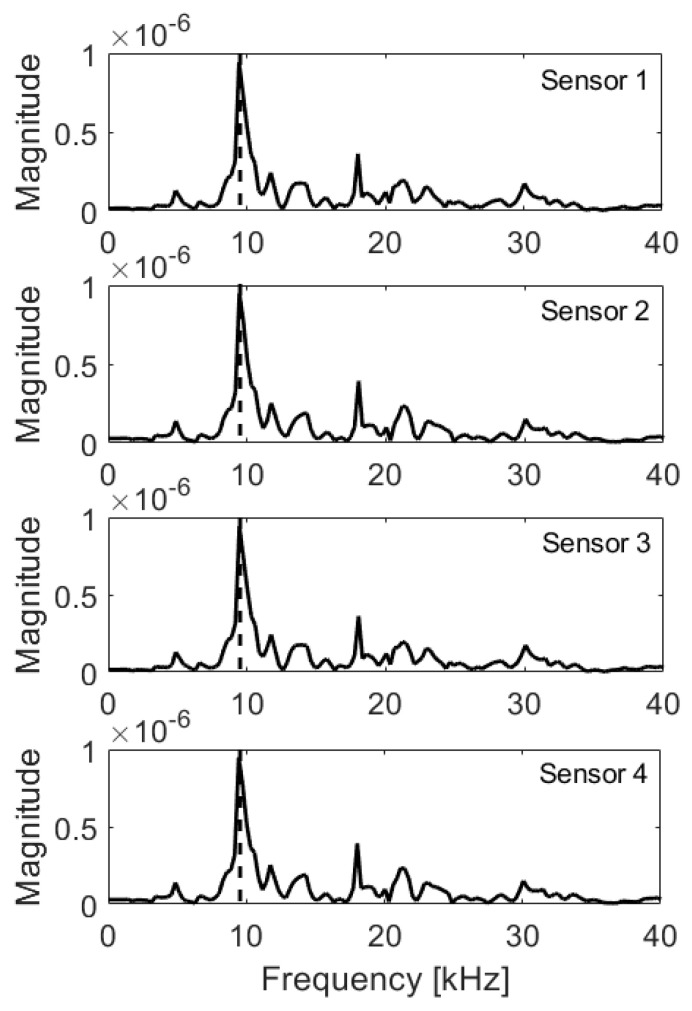
Frequency spectra, solid plate; expected frequency marked with dashed line.

**Figure 7 materials-14-02144-f007:**
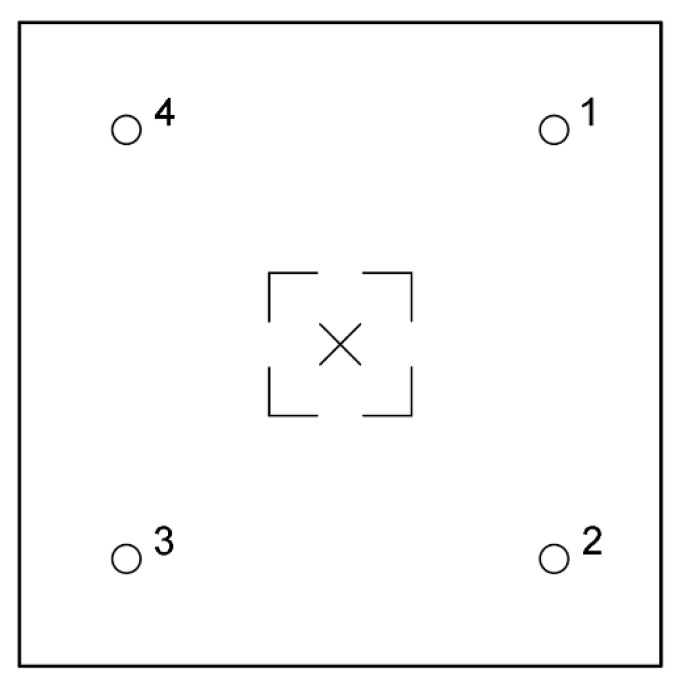
Scheme of void and IE device layout.

**Figure 8 materials-14-02144-f008:**
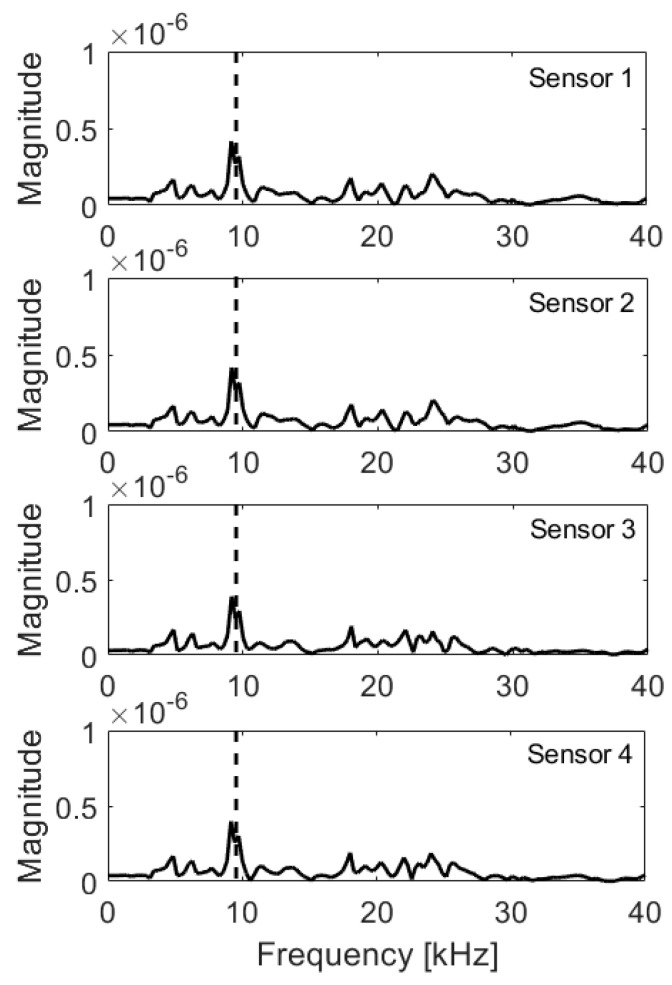
Frequency spectra, void below impactor; expected frequency marked with dashed line.

**Figure 9 materials-14-02144-f009:**
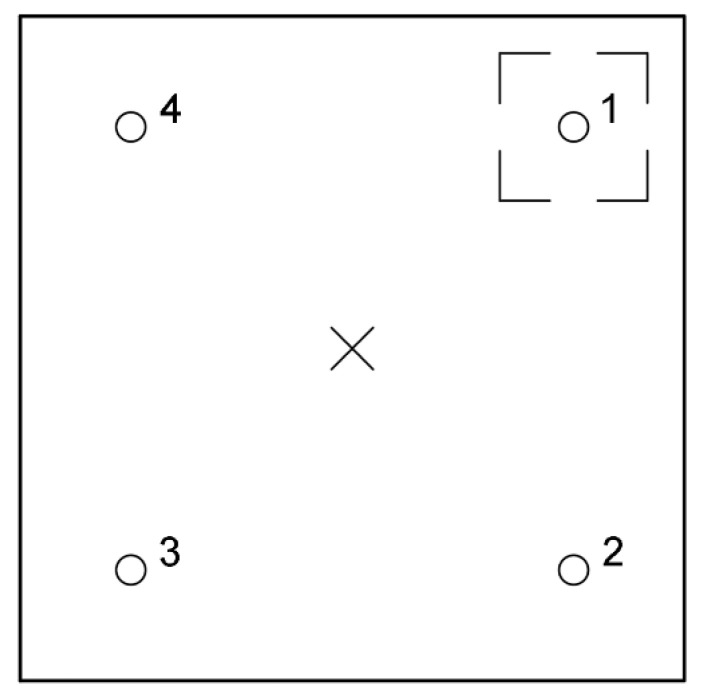
Scheme of void and IE device layout, void below sensor 1.

**Figure 10 materials-14-02144-f010:**
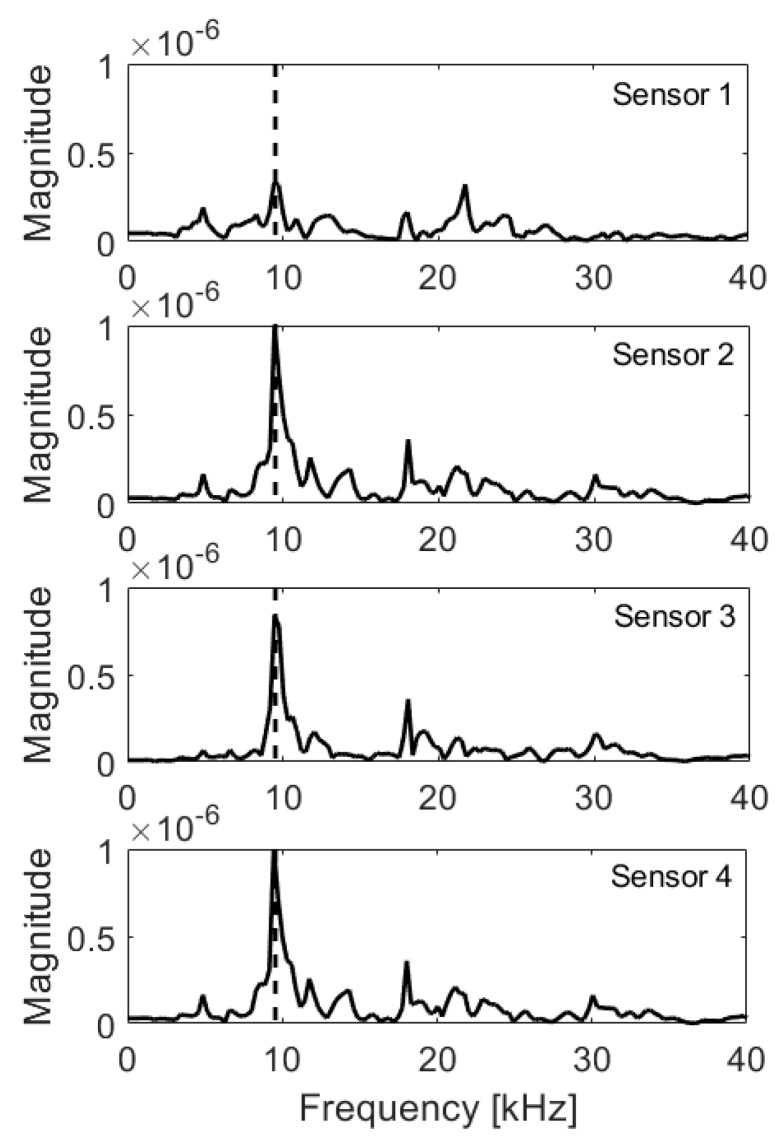
Frequency spectra, void below sensor 1; expected frequency marked with dashed line.

**Figure 11 materials-14-02144-f011:**
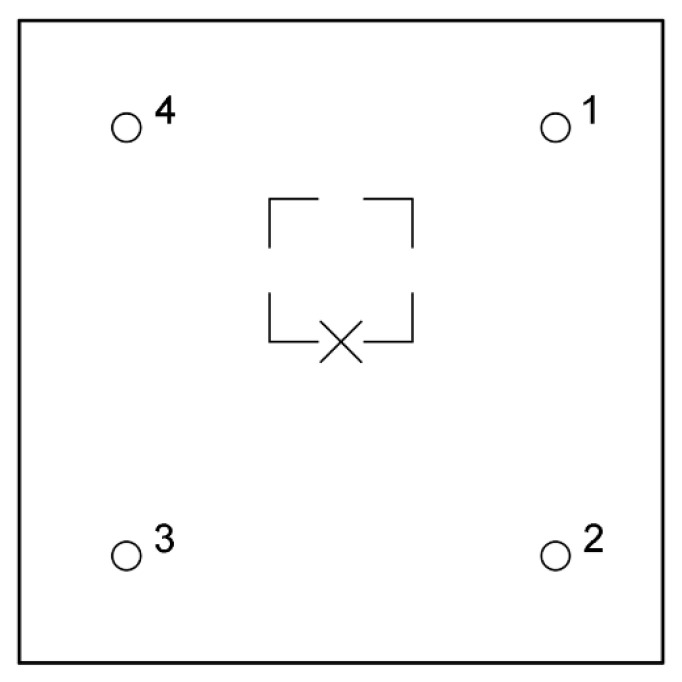
Scheme of void and IE device layout, edge below impactor.

**Figure 12 materials-14-02144-f012:**
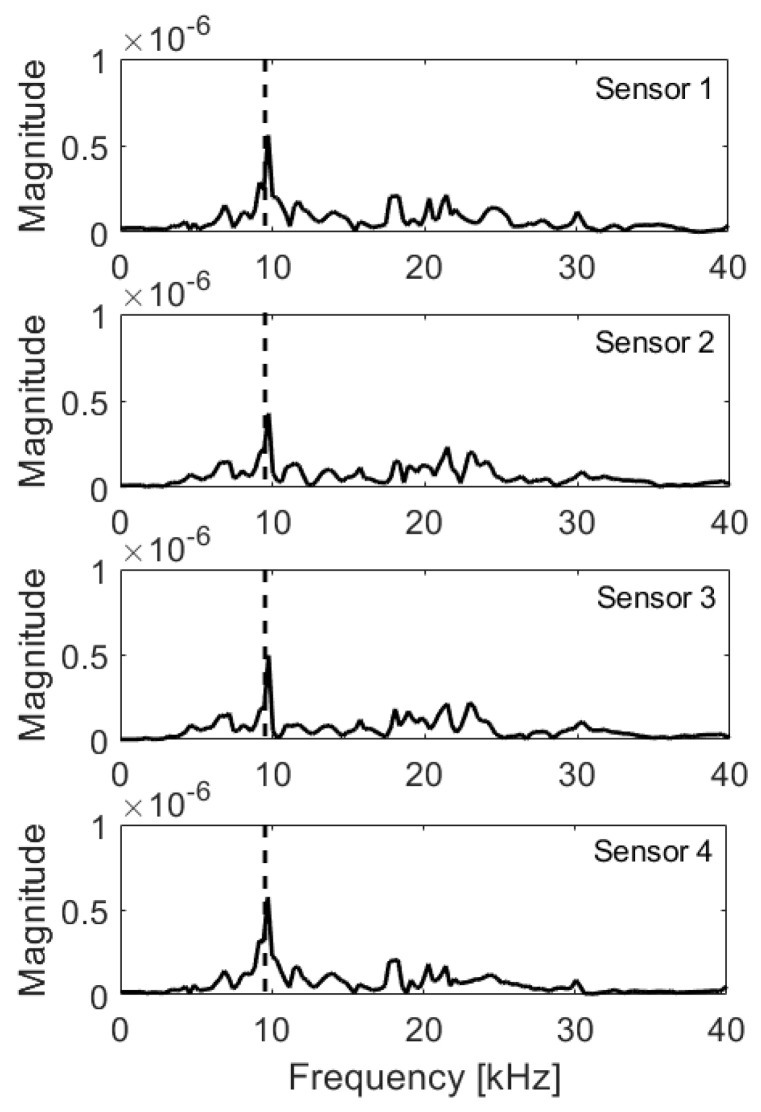
Frequency spectra, edge below impactor; expected frequency marked with dashed line.

**Figure 13 materials-14-02144-f013:**
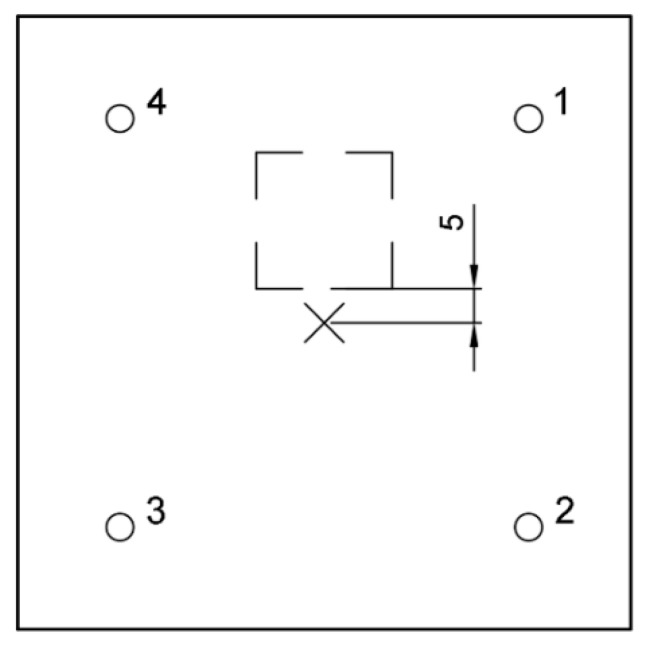
Scheme of void and IE device layout, edge 5 cm from impactor.

**Figure 14 materials-14-02144-f014:**
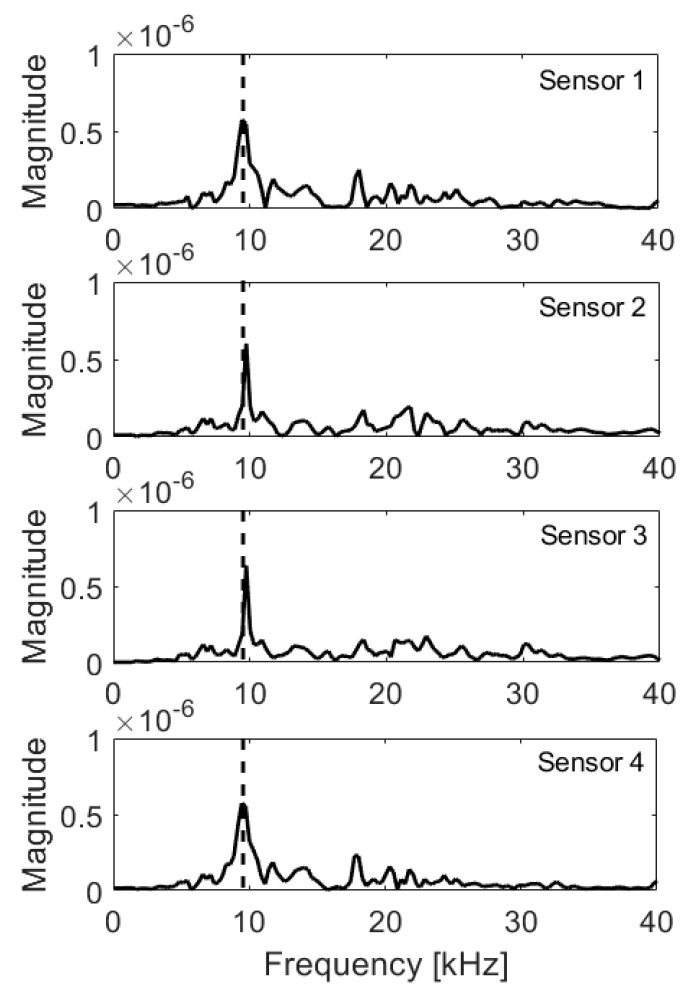
Frequency spectra, edge 5 cm from impactor; expected frequency marked with dashed line.

**Figure 15 materials-14-02144-f015:**
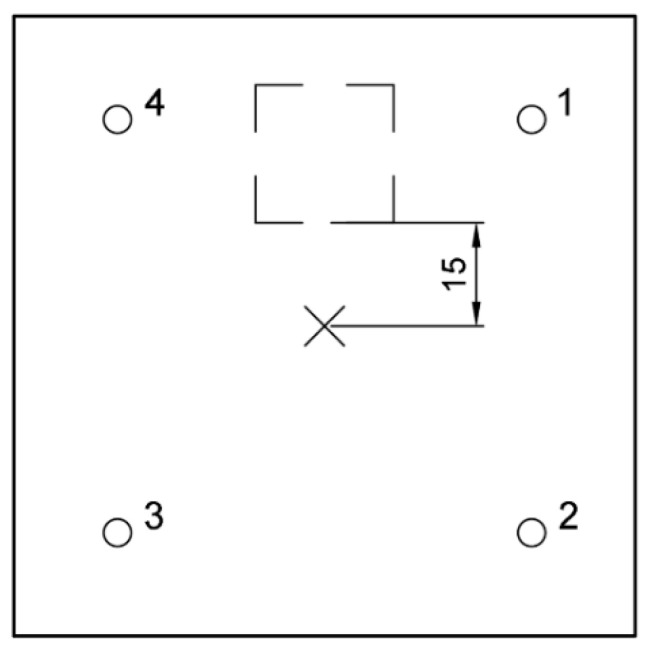
Scheme of void and IE device layout, edge 15 cm from impactor.

**Figure 16 materials-14-02144-f016:**
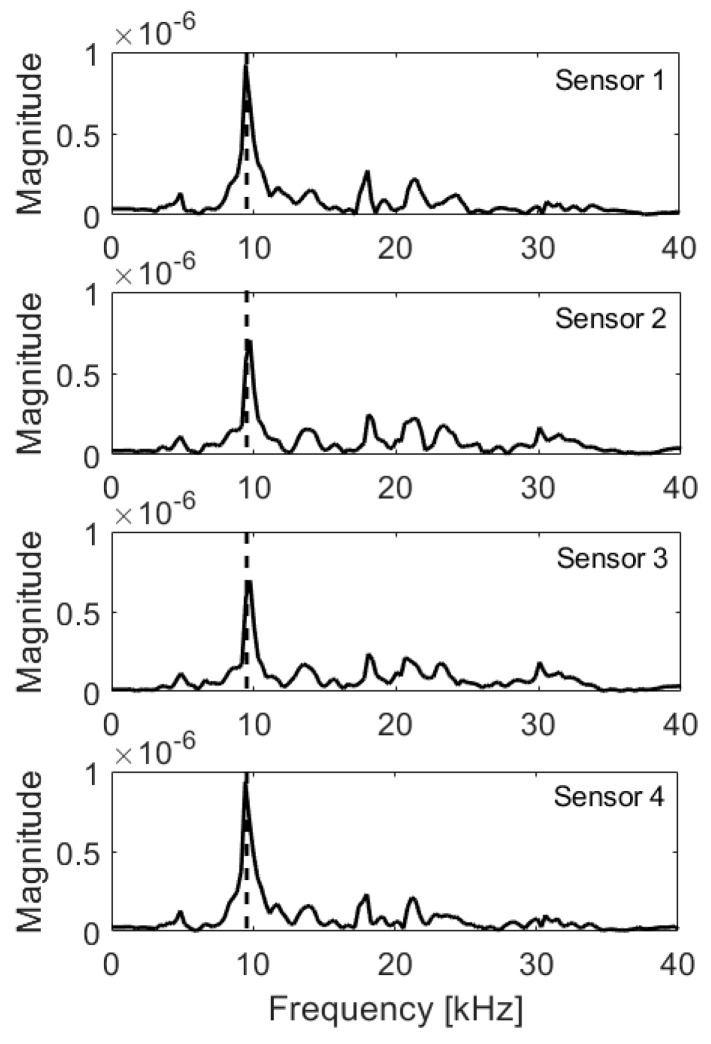
Frequency spectra, edge 15 cm from impactor; expected frequency marked with dashed line.

**Figure 17 materials-14-02144-f017:**
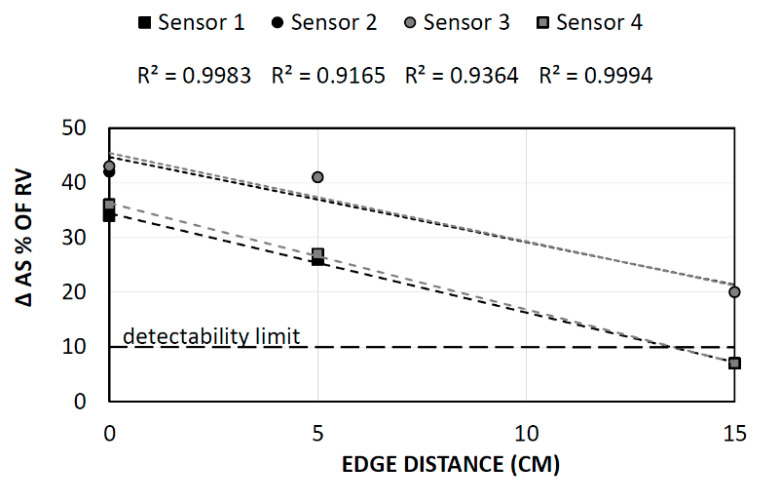
RMS of signal received by each transducer with change of void edge distance from impact point.

**Figure 18 materials-14-02144-f018:**
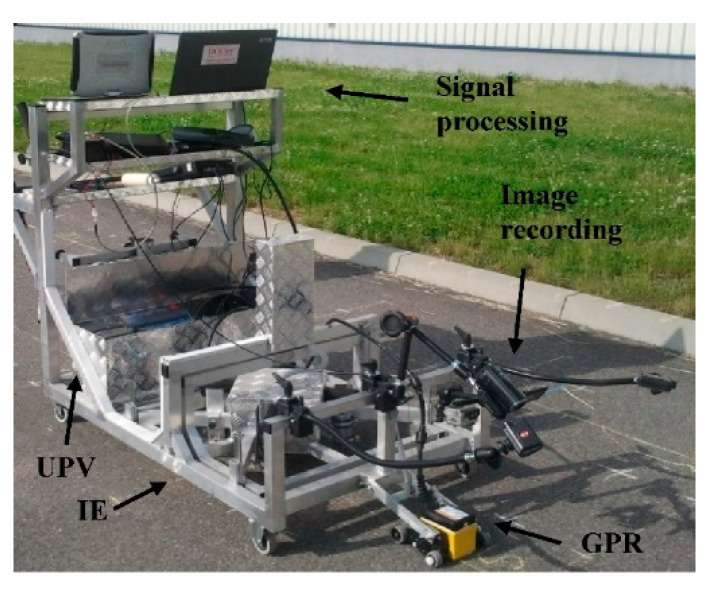
UIR-Scanner (a prototype) for nondestructive testing using complementary NDT methods: GPR (Ground Penetrating Radar), IE (Impact-Echo) and UPV (Ultrasonic Pulse Velocity) equipped with computer systems for signal analysis.

**Figure 19 materials-14-02144-f019:**
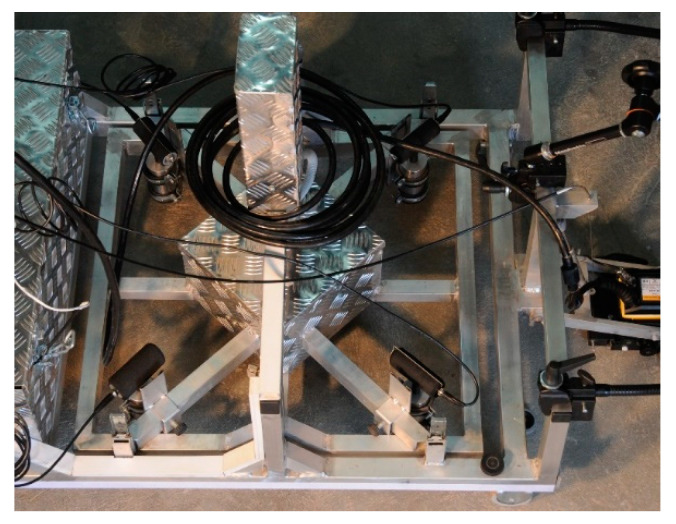
Impact-Echo module integrated into scanner.

**Table 1 materials-14-02144-t001:** Expected peaks for plate with voids (*C_pp_* = 3834 m/s).

Depth [cm]	Frequency [kHz]
9	21.3
11	17.4
20	9.6

**Table 2 materials-14-02144-t002:** Integral and RMS of frequency spectrum, solid plate.

Sensor N°	Integral (10^−6^)	Δ [%]	RMS (10^−7^)	Δ [%]	Detected?
1	3.616	3	1.537	2	N
2	3.521	0	1.515	0	N
3	3.436	2	1.481	2	N
4	3.521	0	1.515	0	N
Mean (RV ^1^)	3.523	-	1.512	-	-

^1^ Reference Value (RV); Detected Y—Yes, N—No, W—Warning.

**Table 3 materials-14-02144-t003:** Integral and RMS of frequency spectrum, void below impactor.

Sensor N°	Integral (10^−6^)	Δ [%]	RMS (10^−7^)	Δ [%]	Detected?
1	2.657	25	0.872	42	Y
2	2.473	30	0.832	45	Y
3	2.331	34	0.791	48	Y
4	2.473	30	0.833	45	Y

Detected Y—Yes, N—No, W—Warning.

**Table 4 materials-14-02144-t004:** Integral and RMS of frequency spectrum, void below sensor 1.

Sensor N°	Integral (10^−6^)	Δ [%]	RMS (10^−7^)	Δ [%]	Detected?
1	2.686	24	0.891	41	Y
2	3.549	1	1.510	0	N
3	2.838	19	1.300	14	Y
4	3.549	1	1.510	0	N

Detected Y—Yes, N—No, W—Warning.

**Table 5 materials-14-02144-t005:** Integral and RMS of frequency spectrum, edge below impactor.

Sensor N°	Integral (10^−6^)	Δ [%]	RMS (10^−7^)	Δ [%]	Detected?
1	2.812	20	0.989	34	Y
2	2.632	25	0.874	42	Y
3	2.445	30	0.858	43	Y
4	2.617	26	0.971	36	Y

Detected Y—Yes, N—No, W—Warning.

**Table 6 materials-14-02144-t006:** Integral and RMS of frequency spectrum, edge below impactor.

Sensor N°	Integral (10^−6^)	Δ [%]	RMS (10^−7^)	Δ [%]	Detected?
1	2.816	20	1.113	26	Y
2	2.490	29	0.886	41	Y
3	2.445	30	0.884	41	Y
4	2.698	23	1.103	27	Y

Detected Y—Yes, N—No, W—Warning.

**Table 7 materials-14-02144-t007:** Integral and RMS of frequency spectrum, edge below impactor.

Sensor N°	Integral (10^−6^)	Δ [%]	RMS (10^−7^)	Δ [%]	Detected?
1	3.230	8	1.407	7	N
2	3.206	9	1.219	20	W
3	3.178	10	1.203	20	Y
4	3.179	10	1.404	7	W

Detected Y—Yes, N—No, W—Warning.

**Table 8 materials-14-02144-t008:** Integral of frequency spectrum (10^−1^), slab with no void.

Sensor N°	Measurement						
	A	B	C	D	E	F	G
1	1.4604	1.7889	1.6837	0.8747	1.4098	1.6977	1.3303
2	1.7374	0.8333	1.6095	1.8303	1.1254	2.3219	0.9677
3	1.0635	1.2686	1.2494	1.1608	1.7251	1.4504	2.2510
4	1.1031	1.9778	1.1016	0.8069	1.1877	1.6299	1.5897

**Table 9 materials-14-02144-t009:** RMS of frequency spectrum (10^−3^), slab with no void.

Sensor N°	Measurement						
	A	B	C	D	E	F	G
1	2.7935	3.4939	3.3806	1.6329	2.7753	3.6157	3.1040
2	3.3197	1.5470	3.0592	3.6199	2.1701	5.0211	1.7858
3	2.6369	3.5801	2.6805	2.4723	3.9014	3.4583	4.8312
4	2.7986	5.0711	2.4122	2.0223	2.7992	3.9092	3.7139

**Table 10 materials-14-02144-t010:** Integral of frequency spectrum (10^−1^), slab with void below impactor.

Sensor N°	Measurement						
	A	B	C	D	E	F	G
1	1.1421	1.0405	0.8709	0.9281	1.5843	0.9358	0.9063
2	1.1993	1.0262	1.5378	1.2899	0.8842	2.0402	1.0260
3	0.8347	1.0381	1.0210	0.9872	0.6384	1.5853	1.1728
4	0.8289	0.8052	0.8035	0.8139	1.2276	0.9977	0.8849

**Table 11 materials-14-02144-t011:** RMS of frequency spectrum (10^−3^), slab with void below impactor.

Sensor N°	Measurement						
	A	B	C	D	E	F	G
1	2.1684	1.8025	1.5266	1.5733	2.8588	1.7611	1.7008
2	2.0785	1.8145	2.7309	2.4415	1.5800	3.7172	1.8479
3	1.5824	2.0308	1.9463	1.7742	1.1389	3.8032	2.4548
4	1.4977	1.4021	1.4673	1.5174	3.0528	2.1219	1.6850

**Table 12 materials-14-02144-t012:** Integral and RMS of frequency spectrum, solid slab.

Sensor N°	Integral (10^−1^)	Δ [%]	RMS (10^−3^)	Δ [%]	Detected?
1	1.464	2	2.971	5	N
2	1.489	4	2.932	6	N
3	1.453	1	3.366	8	N
4	1.342	7	3.247	4	N
Mean (RV ^1^)	1.437	-	3.129	-	-

^1^ Reference Value (RV); Detected Y—Yes, N—No, W—Warning.

**Table 13 materials-14-02144-t013:** Integral and RMS of frequency spectrum, slab with void below impactor.

Sensor N°	Integral (10^−1^)	Δ [%]	RMS (10^−3^)	Δ [%]	Detected?
1	1.058	26	1.913	39	Y
2	1.286	10	2.316	26	Y
3	1.040	28	2.104	33	Y
4	0.909	37	1.821	42	Y

Detected Y—Yes, N—No, W—Warning.

## Data Availability

The data presented in this study are available in [31] (numerical modeling and validation of models) or is available on request from the corresponding author (validation of layout).

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
