# Peer review of "Development of Impact-Echo Multitransducer Device for Automated Concrete Homogeneity Assessment"

_materials, 2021, doi:10.3390/ma14092144_

Round 1
Reviewer 1 Report
The paper is to research a NDT device which is useful in internal status testing of concrete. The results are instructive but the following questions should be noticed.
- In Section 3, more details about the laboratory test and numerical model, such as pictures should be added, so that we can have an intuitive understanding of the model.
Although you wrote the following sentence in Page 3, Line 101-102 “The parametrical studies and verification are not described in this paper for sake of clarity, but were conducted.”, however, necessary details are important because the model validation in Section 3 is the base of following discussion, and unfit parameter setting in numerical models may cause huge errors in results.
- In Page 4, Line 165, the following sentence “Measurements shown that the maximum distance between impactor and receiver giving reliable results is around 90 cm.” should be further explained. More data should be added to support this conclusion. And we are interesting that if this conclusion has universality? Please explain it.
- In Page 6, Line 242, it was concluded that “However, they remain within 5% interval.”, are there more experimental data or theory discussion to further prove it? Because 10% was applied as a fault-detecting criterion in Page 6, Line 243, so the reliability of 5% should be fully ensured.
- In Page 13, Figure 15, a linear relation between received energy and void position was found. However, each line was fitted by only three points, which may cause huge error although calculated R2 is large enough. The position of the measuring points can be properly densified to fit the curve more accurately.
- Page 8, Line 263, “sensor no 3” should be “sensor No.3”.
Author Response
Comments and Suggestions for Authors
The paper is to research a NDT device which is useful in internal status testing of concrete. The results are instructive but the following questions should be noticed.
- In Section 3, more details about the laboratory test and numerical model, such as pictures should be added, so that we can have an intuitive understanding of the model.
Isometrics views of two numerical models used in this study (one for validation and the other one for studying of transducer-void arrangements) were added.
Although you wrote the following sentence in Page 3, Line 101-102 “The parametrical studies and verification are not described in this paper for sake of clarity, but were conducted.”, however, necessary details are important because the model validation in Section 3 is the base of following discussion, and unfit parameter setting in numerical models may cause huge errors in results.
Sensitivity studies were performed on the model by variating material properties and finite element size. The detailed results can be found in Sawicki B., Numerical analyses of stress wave propagation for Impact-Echo testing procedure, Bachelor Thesis, Warsaw University of Technology (2013) (http://dx.doi.org/10.13140/RG.2.2.36643.96801). The Authors decided not to discuss it in detail in the current paper, since it does not bring any new information to the main subject of the paper.
The sentence was rephrased to give better insight in the sensitivity study and the reference to full results added.
- In Page 4, Line 165, the following sentence “Measurements shown that the maximum distance between impactor and receiver giving reliable results is around 90 cm.” should be further explained. More data should be added to support this conclusion. And we are interesting that if this conclusion has universality? Please explain it.
Again, the measurement data is available in Sawicki B., Numerical analyses of stress wave propagation for Impact-Echo testing procedure, Bachelor Thesis, Warsaw University of Technology (2013) and for sake of clarity and brevity of paper is not analyzed in detail here. Furthermore, the exact distance depends on concrete type and impactor size.
Clarifications are added in the text. Figure with three frequency spectra at 10cm, 60cm and 90cm was added.
- In Page 6, Line 242, it was concluded that “However, they remain within 5% interval.”, are there more experimental data or theory discussion to further prove it? Because 10% was applied as a fault-detecting criterion in Page 6, Line 243, so the reliability of 5% should be fully ensured.
This is merely an observation of results, the possible source of variation may come from the numerical noise as explained in the text. The sentence was modified to avoid ambiguity. The 10% limit is established for sake of data analysis and has not physical meaning.
- In Page 13, Figure 15, a linear relation between received energy and void position was found. However, each line was fitted by only three points, which may cause huge error although calculated R2 is large enough. The position of the measuring points can be properly densified to fit the curve more accurately.
Indeed, to find the nonlinear relationship more points would be needed, but in this case, linear relations is rather expected. However, due to small number of points, in the text we do not state that the relationship is linear. Furthermore, this relationship does not influence detectability of fault which is the main goal of simulaitons.
- Page 8, Line 263, “sensor no 3” should be “sensor No.3”.
Corrected
Reviewer 2 Report
I am not able to recommend this paper for publication as it stands.
Therefore, I recommend that a minor revision is warranted, in order to make it much easier for a reader to follow your paper.
I explain my concerns in more detail below.
I ask that the authors specifically address each of my comments in their response.
There are some parts in the text where more or clearer information should be inserted:
- The abstract does not refer to the material being tested and therefore this information should be included.
- The meaning of the UIR scanner acronym is not indicated in any part of the text of the article and therefore this information should be included in the introduction.
- On pag. 4 the reasoning for the definition of the CPP quantity (apparent speed) should be explained, even if reference is made to the references.
- The characteristics of the transducers are not indicated, in particular the frequency. This information should be included, specifying also if the type of probes is always the same.
- The characteristics of the instrumentation used are not indicated (e.g. manufacturer and model) and therefore this information should be included.
- It would be appropriate to enter more details of the test setup (arrangement of the instrumentation). In addition to the photos of the setup including all the instrumentation used (GPV, UPR, IE), a scheme of the entire setup could be inserted for more clearness. Figure 16 could be deleted.
Author Response
Comments and Suggestions for Authors
I am not able to recommend this paper for publication as it stands.
Therefore, I recommend that a minor revision is warranted, in order to make it much easier for a reader to follow your paper.
I explain my concerns in more detail below.
I ask that the authors specifically address each of my comments in their response.
There are some parts in the text where more or clearer information should be inserted:
- The abstract does not refer to the material being tested and therefore this information should be included.
Specification of material (concrete) added in the first sentence of abstract
- The meaning of the UIR scanner acronym is not indicated in any part of the text of the article and therefore this information should be included in the introduction.
In introduction formatting of techniques’ names has been modified to clarify this acronym (Ultrasonic, Impact-echo, Radar)
- On pag. 4 the reasoning for the definition of the CPP quantity (apparent speed) should be explained, even if reference is made to the references.
Specification of reason of this apparent velocity (Lamb waves) added.
The authors prefer not to discuss this aspect in details for sake of clarity of the paper. An interested reader can refer to the literature
- The characteristics of the transducers are not indicated, in particular the frequency. This information should be included, specifying also if the type of probes is always the same.
Specification of type of transducer added.
All measurements were taken by DOCter Mark IV transducers from Germann Instruments. Unfortunately the manual of device does not contain precise characteristics of transducers.
- The characteristics of the instrumentation used are not indicated (e.g. manufacturer and model) and therefore this information should be included.
As above.
- It would be appropriate to enter more details of the test setup (arrangement of the instrumentation). In addition to the photos of the setup including all the instrumentation used (GPV, UPR, IE), a scheme of the entire setup could be inserted for more clearness. Figure 16 could be deleted.
In the current paper, neither UPV nor GPR were used and therefore they are not discussed. The fig. 16 presents the scanner with IE module implemented illustrating fruits of the development procedure discussed in this paper. Therefore, for sake of clarity, IE module itself is presented in fig. 17.
Reviewer 3 Report
The manuscript presents the Development of Impact-Echo multitransducer device for automated concrete homogeneity assessment.
The research area covered by the paper can be of potential very interest for the readers of this Journal of Materials. Nevertheless, in reviewer’s opinion, the manuscript has deficiencies that prevent its publication.
The work and research builds on the authors' previous research.
The introduction chapter needs to be fundamentally reworked. There is extensive research in the area. There are many non-destructive methods (sensors) for diagnosing building structures. There is to state a need for clear motivation and potential benefits of the presented research.
Potentially interesting references from the research area include:
Pazdera, L., et. al. Measurement and utilization of acoustic emission for the analysis and monitoring of concrete slabs on the Subsoil.Periodica Polytechnica Civil Engineering 2019, 63 (2), 608-620.
Huo, L. et. al. Bond-Slip Monitoring of Concrete Structures Using Smart Sensors—A Review. Sensors 2019, 19, 1231.
Figure 1 is very small. Magnify. Who owns the copyright?
It is necessary to provide more details about the created calculation model and calculation parameters. (Part - 3.2 Numerical model, ......)
Is there a reference to [35]? DOI? www?
All references are in the wrong format. I challenged the authors to work out a revision of the manuscript with more interest and control according to the MDPI template.
Extend the details in the part: (4. Experimental studies on a large span floor)
Figure 4. enlarge the image.
Figure 6. enlarge the image.
Figure 8. enlarge the image.
Figure 10. enlarge the image.
Figure 12. enlarge the image.
Part of the Discussion is missing in the manuscript. The achieved results must be put into the overall context and current knowledge. Any restrictions should also be mentioned.
The conclusion needs to be fundamentally reworked. It's just a summary of information.
The manuscript is well structured but has a particularly only descriptive character. There is a lack of criticism (recommendation) throughout the analysis of the information in the manuscript.
Overall, the topic and the research area is interesting, but the author's must rework the manuscript and improve the overall informational value and presentation of new knowledge.
Author Response
Comments and Suggestions for Authors
The manuscript presents the Development of Impact-Echo multitransducer device for automated concrete homogeneity assessment.
The research area covered by the paper can be of potential very interest for the readers of this Journal of Materials. Nevertheless, in reviewer’s opinion, the manuscript has deficiencies that prevent its publication.
The work and research builds on the authors' previous research.
The introduction chapter needs to be fundamentally reworked. There is extensive research in the area. There are many non-destructive methods (sensors) for diagnosing building structures. There is to state a need for clear motivation and potential benefits of the presented research.
Potentially interesting references from the research area include:
Pazdera, L., et. al. Measurement and utilization of acoustic emission for the analysis and monitoring of concrete slabs on the Subsoil.Periodica Polytechnica Civil Engineering 2019, 63 (2), 608-620.
Huo, L. et. al. Bond-Slip Monitoring of Concrete Structures Using Smart Sensors—A Review. Sensors 2019, 19, 1231.
The Authors agree that there are many NDTs available, however the aim of the current paper is not to review or compare them. Introduction is only supposed to state the motivation behind work presented here. i.e. development and application of Impact Echo module for UIR-Scanner developed at Warsaw University of Technology. The choice of three measurement methods in the scanner (Ultrasonic, Geopenetrating radar and Impact-Echo) was done on basis of their complementarity. Therefore IE is not supposed to compete or replace any of these methods.
Figure 1 is very small. Magnify. Who owns the copyright?
Figure 1 was supposed to be modified by Editors before send the paper to the revision. The current Fig.1 was not published before and was enlarged.
It is necessary to provide more details about the created calculation model and calculation parameters. (Part - 3.2 Numerical model, ......)
More details about model were added
Is there a reference to [35]? DOI? www?
DOI was added
All references are in the wrong format. I challenged the authors to work out a revision of the manuscript with more interest and control according to the MDPI template.
References were modified accordingly
Manuscript was checked for accordance with the MDPI template.
Extend the details in the part: (4. Experimental studies on a large span floor)
Additional comments on the results added.
The results are presented in Sawicki B., Numerical analyses of stress wave propagation for Impact-Echo testing procedure, Bachelor Thesis, Warsaw University of Technology (2013) (http://dx.doi.org/10.13140/RG.2.2.36643.96801) as stated at the end of section 4. Experimental studies on a large span floor. In opinion of Authors it is not necessary to discuss them in detail here – they do not bring any new information on performance of the proposed IE module, and are described to merely show the development procedure of the module.
Figure 4. enlarge the image.
Figure 6. enlarge the image.
Figure 8. enlarge the image.
Figure 10. enlarge the image.
Figure 12. enlarge the image.
All relevant figures were enlarged
Part of the Discussion is missing in the manuscript. The achieved results must be put into the overall context and current knowledge. Any restrictions should also be mentioned.
The conclusion needs to be fundamentally reworked. It's just a summary of information.
Conclusions modified appropriately.
The manuscript is well structured but has a particularly only descriptive character. There is a lack of criticism (recommendation) throughout the analysis of the information in the manuscript.
Discussion on limitations of the presented approach and IE method itself is added in section 6.4
Overall, the topic and the research area is interesting, but the author's must rework the manuscript and improve the overall informational value and presentation of new knowledge.
Round 2
Reviewer 3 Report
The research area and results are from the context of the manuscript can better understand.
Thanks for the comments and manuscript edits.
The manuscript can be accepted for publication.